# Classification of Hyperspectral Image Based on Double-Branch Dual-Attention Mechanism Network

**Rui Li** [1],*, **Shunyi Zheng** [1], **Chenxi Duan** [2], **Yang Yang** [1] and **Xiqi Wang** [1]

[1]  School of Remote Sensing and Information Engineering, Wuhan University, Wuhan 430079, China; syzheng@whu.edu.cn (S.Z.); yangyang001@whu.edu.cn (Y.Y.); wangxiqi@whu.edu.cn (X.W.)

[2]  State Key Laboratory of Information Engineering in Surveying, Mapping, and Remote Sensing, Wuhan University, Wuhan 430079, China; chenxiduan@whu.edu.cn

*  Correspondence: lironui@whu.edu.cn

**Abstract:** In recent years, researchers have paid increasing attention on hyperspectral image (HSI) classification using deep learning methods. To improve the accuracy and reduce the training samples, we propose a double-branch dual-attention mechanism network (DBDA) for HSI classification in this paper. Two branches are designed in DBDA to capture plenty of spectral and spatial features contained in HSI. Furthermore, a channel attention block and a spatial attention block are applied to these two branches respectively, which enables DBDA to refine and optimize the extracted feature maps. A series of experiments on four hyperspectral datasets show that the proposed framework has superior performance to the state-of-the-art algorithm, especially when the training samples are signally lacking.

**Keywords:** hyperspectral image classification; deep learning; channel-wise attention mechanism; spatial-wise attention mechanism

## 1. Introduction

Remote sensing images can be categorized by their spatial, spectral, and temporal resolutions [1], and has been generally researched for many areas such as land-cover mapping [2], water monitoring [3], and anomaly detection [4]. As a particular type of remote sensing images with high spectral resolution, hyperspectral image (HSI) contains plentiful information both in the spectral and spatial dimension [5]. HSI has been used in many fields including vegetation cover monitoring [6], atmospheric environmental research [7], and change area detection [8], among others. Supervised classification is an essential task of HSI, and is the common technology used in the above applications. However, the over-redundancy of spectral band information and limited training samples account for a huge challenge to HSI classification.

Early spectral-based attempts including support vector machines (SVM) [9], multinomial logistic regression (MLR) [10,11], and random or dynamic subspace [12,13], focus on the spectral characteristics of HSI. Nevertheless, another useful piece of information is that the adjacent pixels are possibly of the same category, but the spectral-based methods ignore the high spatial correlation and local consistency of HSI. Therefore, the increasing number of classification frameworks based on spectral-spatial features have been presented. Two types of low-level features, morphological profiles [14] and Gabor feature [15], were designed to represent the spatial information. Based on SVM, the morphological kernel [16] and the composite kernel [17] methods were also proposed to exploit spectral-spatial information. Although above attempts improve the accuracy of the classifier, these methods highly depend on the hand-crafted descriptors.

Deep learning (DL) has shown powerful capabilities in automatically extracting nonlinear and hierarchical features. A great surge of computer vision tasks have benefited from DL and made



significant breakthroughs, such as objection detection [18], natural language processing [19], and image classification [20]. As a typical classification tasks, HSI classification has been deeply influenced by DL and has obtained excellent improvements.

In [21], Chen introduced stacked autoencoders (SAE) for extracting useful features. Similarly, Tao [22] used two sparse SAEs to capture spectral and spatial information separately. Ma et al. [23] proposed an updated deep auto-encoder (DAE) to extract spectral-spatial features, and designed a novel synergic representation to handle the small-scale training set. Zhang et al. [24] used a recursive autoencoder (RAE) to extracted high-level features from the neighborhoods of the target pixel and used a new weighting scheme to fuse the spatial information. In [25], Chen et al. proposed a classification method based on deep belief network (DBN) and restricted Boltzmann machine (RBM).

However, in the above-mentioned methods, the input is one-dimensional. Although the spatial information is utilized, the initial structure is destroyed. Since convolutional neural networks (CNN) could exploit spatial feature while retaining the original structure, some novel solutions have been introduced with the advent of CNN. Zhao et al. [26] adopted CNN as a feature extractor in their framework. Lee et al. [27] proposed a contextual deep CNN (CDCNN) with deeper and wider networks. In [28], Chen et al. designed 3D-CNN-based feature extractor model integrated with regularization.

Although DL has brought promising improvements in HSI classification, the demand of DL for training samples is enormous, while the cost of manual annotation is rather expensive for HSI. Generally, deeper networks can capture finer features, but it will be harder to train deeper networks. The emergence of the residual network (ResNet) [29] and the dense convolutional network (DenseNet) [30] eases the difficulty of training of deeper networks. Inspired by the ResNet, Zhong et al. [31] proposed a spectral-spatial residual network (SSRN), which is more effective with limited training samples. Wang et al. [32] introduced DenseNet to their fast dense spectral-spatial convolution (FDSSC) algorithm.

To optimize the discrimination of extracted features, the attention mechanism was adopted to refine the feature maps. Fang et al. [33] designed a 3-D dense convolutional network with spectral-wise attention mechanism (MSDN-SA) based on DenseNet and attention mechanism. Ma et al. [34] proposed a double-branch multi-attention mechanism network (DBMA) motivated by the convolutional block attention module (CBAM) [35], and obtained the best classification results.

Inspired by the latest development of DL fields, some new methods could be observed in the literature. Mou et al. [36] proposed a recurrent neural networks (RNN) framework for HSI classification in which hyperspectral pixels were analyzed via the sequential perspective. Because of the severe absence of labelled samples in HSI, semi-supervised learning (SSL) [37], generative adversarial network (GAN) [38], and active learning (AL) [39] were introduced to alleviate this problem. In [40], spectral-spatial capsule networks (CapsNets) were designed to weaken the complexity of the network and enhance the accuracy of the classification. Furthermore, self-pace learning [41], self-taught learning [42], and superpixel-based methods [43] are also worth noting.

In this paper, inspired by the state-of-the-art DBMA algorithm and an adaptive self-attention mechanism dual attention network (DANet) [44], we design the double-branch dual-attention mechanism network (DBDA) for HSI classification. The proposed framework contains two branches named the spectral branch and spatial branch, which capture spectral and spatial features separately. The channel-wise attention mechanism and spatial-wise attention mechanism are adopted to refine the feature maps. By concatenating the output of the two branches, we obtain syncretic spectral-spatial features. Finally, the classification results are determined using a softmax function. The three significant contributions of this paper could be listed as follows:

- Based on DenseNet and 3D-CNN, we propose an end-to-end framework double-branch dual-attention mechanism network (DBDA). The spectral branch and spatial branch of the proposed framework can exploit features respectively without any feature engineering.
- A flexible and adaptive self-attention mechanism is introduced to both the spectral and spatial dimensions. The channel-wise attention block is designed to focus on the information-rich spectral bands, and the spatial-wise attention block is built to concentrate on the information-rich pixels.

- The DBDA obtains the state-of-the-art classification accuracy in four datasets with limited training data. Furthermore, the time consumption of our proposed network is less than the two compared deep-learning algorithms.

The rest of this paper is arranged as follows: In Section 2, we illustrate the related work briefly. The detailed structure of DBDA is given in Section 3. In Sections 4 and 5, we provide and analyze the experimental results. Finally, a conclusion of the entire paper with a direction for future work is presented in Section 6.

All of our code is available publicly at https://github.com/lironui/Double-Branch-Dual-Attention-Mechanism-Network.

## 2. Related Work

In this section, we are going to make a brief introduction to the basic modules used in DBDA, including the 3D-cube-based HSI classification framework, 3D-CNN with batch normalization, ResNet and DenseNet, the channel-wise attention mechanism, and the spatial-wise attention mechanism. Since both the number of the HSI spectrums and convolutional kernels could be referred to as channels, we call the number of the HSI spectrums ***bands***, and named the number of the convolutional kernels ***channels*** to avoid confusion.

### 2.1. HSI Classification Framework Based on 3D-Cube

Unlike traditional pixel-based methods that only use spectral features, 3D-cube-based methods like SSRN [31], FDSSC [32], DBMA [34], and our proposed framework exploit both spectral and spatial information. The pixel-based methods use the pixel individually to train the network, but the 3D-cube-based methods take the target pixel and its adjacent pixels as input. Certainly, the labels of adjacent central pixels are not fed into the network, and we only explore the abundant spatial information around the target pixel. Generally, the difference between pixel-based methods and 3D-cube-based methods is the input size of the former is $1 \times 1 \times b$, while that of the latter is $p \times p \times b$, where $p \times p$ represents the number of neighboring pixels and $b$ denotes the number of spectral bands.

### 2.2. 3D-CNN with Batch Normalization

3D-CNN with batch normalization (BN) [45] is a common element in 3D-cube-based deep learning models. Inputting abundant labelled images, deep learning models with multiple nonlinear layers can learn hierarchical representations, and the multilevel convolutional layers empower CNN to learn characteristics under sparsity constraint more discriminatively. 1D-CNN and 2D-CNN only use spectral features or capture local spatial features of the pixels. When classifying HSI that contains plenty of both spatial and spectral information, 3D-CNN should be adopted to get reasonable results. Therefore, we use 3D-CNN as the basic structure of the DBDA. Moreover, we add a BN layer in each 3D-CNN layer to improve the numerical stability.

As shown in Figure 1, with $n_m$ input feature maps at the size of $p_m \times p_m \times b_m$, a 3D-CNN layer contains $k_{m+1}$ channels in the size of $\alpha_{m+1} \times \alpha_{m+1} \times d_{m+1}$, which generates the $n_{m+1}$ output feature maps of size $p_{m+1} \times p_{m+1} \times b_{m+1}$. The $i$th output of the $(m+1)$th 3D-CNN layer with BN could be calculated as:

$$X_i^{m+1} = \text{R}\left(\sum\nolimits_{j=1}^{n_m} \widehat{X}_j^m * H_i^{m+1} + b_i^{m+1}\right) \tag{1}$$

$$\widehat{X}^m = \frac{X^m - E(X^m)}{Var(X^m)} \tag{2}$$

in which $X_j^m \in \mathbb{R}^{p \times p \times b}$ is the $j$th input feature map of the $(m+1)$th layer, and $\widehat{X}^m$ is the output after the BN in the $m$th layer. $E(\cdot)$ and $Var(\cdot)$ denote the expectation and variance function of the input separately. $H_i^{m+1}$ and $b_i^{m+1}$ represent the weights and biases of the $(m+1)$th 3D-CNN layer, $*$ is the

3D convolutional operation, and $R(\cdot)$ denotes the activation function that introduces the nonlinear unit of the network.

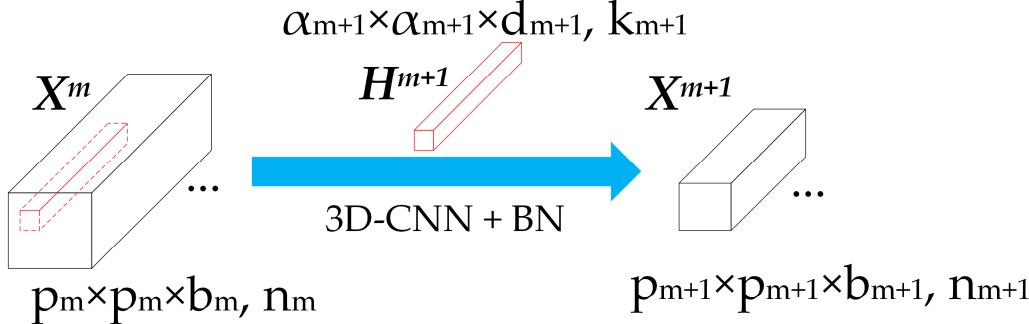

**Figure 1.** The structure of 3D-convolutional neural networks (CNN) with a batch normalization (BN) layer.

*2.3. ResNet and DenseNet*

Normally, the more convolutional layers, the better a network will perform. However, too many layers may make the problems of vanishing and exploding gradients worse. ResNet [29] and DenseNet [30] are valid and efficient methods to escape this dilemma.

Generally, a skip connection is added to the conventional CNN model in ResNet. As indicated in Figure 2a, $H$ denotes hidden block, which is a module containing convolutional layers, activation layers, and BN layers. The skip connection, which could be regarded as an identity mapping, enables the input data to pass directly through the network. The residual block is the basic unit in ResNet, and the output of the $l$th residual block can be calculated as:

$$x_l = H_l(x_{l-1}) + x_{l-1} \tag{3}$$

Based on ResNet, DenseNet connects all layers directly to ensure maximum information flow between each layer of the network. Instead of combining features through summation like ResNet, DenseNet combines features via concatenating them in the channel dimension. The dense block is the basic unit in DenseNet, and the output of the $l$th dense block can be computed as:

$$x_l = H_l[x_0, \ x_1, \ldots, x_{l-1}] \tag{4}$$

in which $H_l$ is a module including convolution layers, activation layers, and BN layers, and $x_0, \ x_1, \ldots, x_{l-1}$ denote the feature maps generated by the preceding dense blocks. As shown in Figure 2b, more connections ensure more information flow in the DenseNet. Specifically, DenseNet with $L$ layers owns $L(L+1)/2$, while traditional convolutional networks with equal layers only have $L$ direct connections.

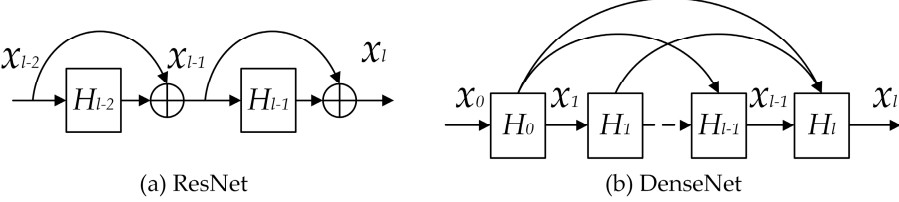

(a) ResNet          (b) DenseNet

**Figure 2.** The architecture of residual network (ResNet) and dense convolutional network (DenseNet).

The structure of the dense connection block used in our framework can be seen in Figure 3. The Mish in Figure 3 means the activation function adopted in our framework, and the details about Mish can be seen in Section 3.2.1. Supposing that the shape of the input feature maps is $p \times p \times b$ with $n$

channels, and that each convolution layer is composed of $k$ kernels in the shape of $1 \times 1 \times d$, then each layer generates feature maps in the shape of $p \times p \times b$ with $k$ channels. However, a dense connection concatenates feature maps at the channel dimension, so there is a linear relationship between the number of channels and the number of convolution layers. The output with $k_m$ channels generated by an $m$-layers dense block can be formulated as:

$$k_m = b + (m-1) \times k \tag{5}$$

where $b$ represents the channel's number in the input feature maps.

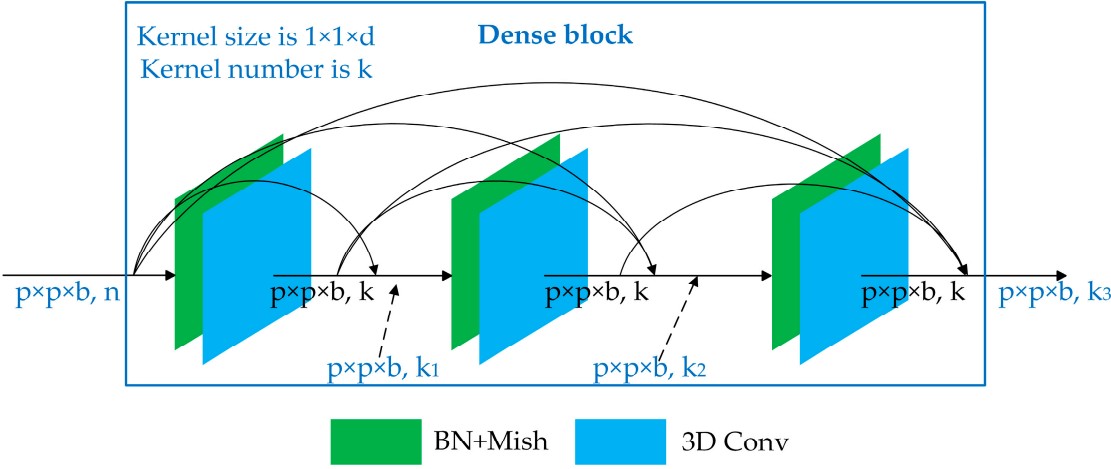

**Figure 3.** The structure of the dense block used in our framework.

## *2.4. Attention Mechanism*

A shortcoming of the 3D-CNN is that all the spatial pixels and spectral bands own the equivalent weights in the spatial and spectral domains. Obviously, different spectral bands and spatial pixels make different contributions to extracting features. The attention mechanism is a powerful technique to deal with this problem. Motivated by the human visual perception process [46], the attention mechanism is designed to focus more on the informative areas and takes less account of non-essential areas. The attention mechanism has been used for image categorization [47] and was later proved to be outstanding in other areas including image caption [48], text to image synthesis [49] and scene segmentation [44], etc. In DANet [44], the channel attention block and spatial attention block can be adopted to increase the weight of compelling channels and pixels. The two blocks will be introduced in detail as the following.

### 2.4.1. Spectral Attention Block

As illustrated in Figure 4a, the channel attention map $X \in \mathbb{R}^{c \times c}$ is directly computed from the initial input $A \in \mathbb{R}^{c \times p \times p}$, where $p \times p$ is the patch size of the input, and $c$ denotes the number of the input channels. Concretely, a matrix multiplication between $A$ and $A^{\mathrm{T}}$ is operated, and to obtain the channel attention map $X \in \mathbb{R}^{c \times c}$, a softmax layer is connected as:

$$x_{ji} = \frac{exp\left(A_i \times A_j\right)}{\sum_{i=1}^{C} exp\left(A_i \times A_j\right)} \tag{6}$$

in which $x_{ji}$ means the $i$th channel's influence on the $j$th channel. Then, the results of matrix multiplication between $X^T$ and $A$ are reshaped into $\mathbb{R}^{c \times p \times p}$. Finally, the reshaped results are weighted by a parameter of scale $\alpha$ and added input $A$ to acquire the final spectral attention map $E \in \mathbb{R}^{c \times p \times p}$:

$$E_j = \alpha \sum_{i=1}^{C} (x_{ji} A_j) + A_j \tag{7}$$

where $\alpha$ is initialized as zero and can be learned gradually. The final map $E$ encompasses the weighted summations of all channels' features, which can describe long-range dependencies and boost the discriminability about features.

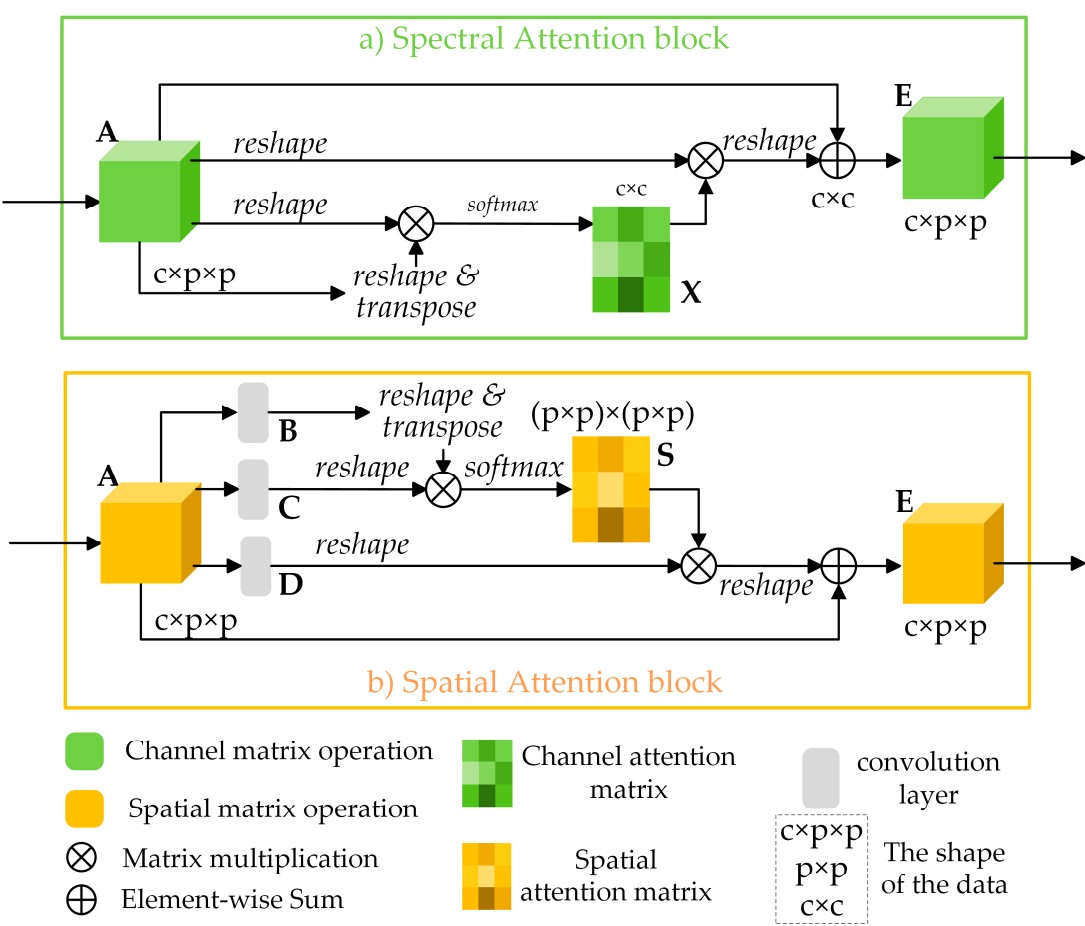

**Figure 4.** The details of the spectral attention block and the spatial attention block.

2.4.2. Spatial Attention Block

As illustrated in Figure 4b, given an input feature map $A \in \mathbb{R}^{c \times p \times p}$, two convolution layers are adopted to generate new feature maps $B$ and $C$ respectively, where $\{B, C\} \in \mathbb{R}^{c \times p \times p}$. Next, $B$ and $C$ are reshaped into $\mathbb{R}^{c \times n}$, where $n = p \times p$ is the number of pixels. Then a multiplication of matrices is executed between $B$ and $C$, and a softmax layer is attached subsequently to calculate the spatial attention feature maps $S \in \mathbb{R}^{n \times n}$:

$$s_{ji} = \frac{exp(B_i + C_j)}{\sum_{i=1}^{N} exp(B_i + C_j)} \tag{8}$$

where $s_{ji}$ measures the impact of $i$th pixel to the $j$th pixel. The closer feature representations of the two pixels signify a stronger correlation between them.

The initial input feature $A$ is simultaneously fed into a convolution layer to obtain a new feature map $D \in \mathbb{R}^{c \times p \times p}$ which is reshaped into $\mathbb{R}^{c \times n}$ subsequently. Then a multiplication of matrices is performed between $D$ and $S^T$, and the result is reshaped into $\mathbb{R}^{c \times p \times p}$ as:

$$E_j = \beta \sum_{i=1}^{N} (s_{ji} D_j) + A_j \tag{9}$$

where $\beta$ with a zero initial value can be learned to assign more weight gradually. By Equation (9), it can be inferred that all positions and original features are added with a certain weight to get the final feature $E \in \mathbb{R}^{c \times p \times p}$. Therefore, long-range contextual information in the spatial dimension is modeled as $E$.

## 3. Methodology

The procedure of the DBDA framework contains three steps: dataset generation, training and validation, and prediction. Figure 5 illustrates the whole framework of our method.

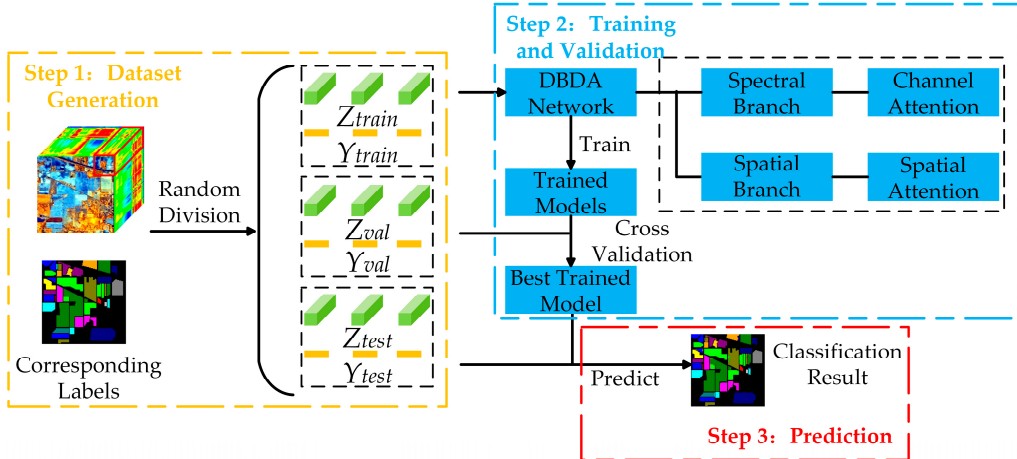

**Figure 5.** The procedure of our proposed double-branch dual-attention (DBDA) framework.

An HSI dataset $X$ is supposed to be composed of $N$ labelled pixels $\{x_1, x_2, \ldots, x_n\} \in \mathbb{R}^{1 \times 1 \times b}$, where $b$ represents the bands, and the corresponding category label set is $Y = \{y_1, y_2, \ldots, y_n\} \in \mathbb{R}^{1 \times 1 \times c}$, where $c$ denotes the numbers of land cover classes.

In the dataset generation step, $p \times p$ neighboring pixels of the center pixel $x_i$ is selected from the original data to generate the 3D-cubes set $\{z_1, z_2, \ldots, z_n\} \in \mathbb{R}^{p \times p \times b}$. If the target pixel is on the edge of the image, the values of missing adjacent pixels are set as zero. The $p$, i.e., patch size, is set as 9 in our framework. Then, the 3D-cubes set is randomly divided into training set $Z_{train}$, validation set $Z_{val}$, and testing set $Z_{test}$. Accordingly, their corresponding label vectors are divided into $Y_{train}$, $Y_{val}$, and $Y_{test}$. Certainly, the labels of neighboring pixels are not visible to the network, we use the spatial information around target pixel only.

In the training and validation steps, the training set is used to update the parameters for many epochs, while the validation set is adopted to monitor the performance of models and to select the best-trained model.

In the prediction step, the test set is chosen to verify the effectiveness of the trained model.

The commonly used quantitative indexes for HSI classification to measure the difference between predicted results and real values is the cross-entropy loss function, which is defined as

$$C(\widehat{y}, y) = \sum_{m=1}^{L} y_m \left( log \sum_{n=1}^{L} e^{\widehat{y_n}} - \widehat{y_m} \right) \tag{10}$$

where $\widehat{y} = [\widehat{y_1}, \widehat{y_2}, \ldots, \widehat{y_L}]$ means the label vector predicted by the model and $y = [y_1, y_2, \ldots y_L]$ represents the ground-truth label vector.

### 3.1. The Framework of the DBDA Network

The whole structure of the DBDA network can be seen in Figure 6. For convenience, we call the top branch *Spectral Branch* and name the bottom branch *Spatial Branch*. The input is fed into spectral branch and spatial branch respectively to get the spectral feature maps and spatial feature maps. Then the fusion operation between spectral and spatial feature maps are adopted to get the classification results.

The following parts introduce the spectral branch, Spatial Branch and spectral and spatial fusion operation taking the Indian Pines (IP) dataset as an example; the patch size is assigned as $9 \times 9 \times 200$. To facilitate the understanding for the matrices mentioned below such as $(9 \times 9 \times 97, \ 24)$, the $9 \times 9 \times 97$ represent the height, width, and depth of the 3D-cube, and 24 represents the number of 3D-cubes generated by 3D-CNN.

The IP dataset contains $145 \times 145$ pixels with 200 spectral bands, that is, the size of IP is $145 \times 145 \times 200$. The details of IP can be seen in Table 3. There are only 10, 249 pixels have corresponding labels, and the other pixels are background.

#### 3.1.1. Spectral Branch with the Channel Attention Block

First, a 3D-CNN layer with a $1 \times 1 \times 7$ kernel size is used. The down sampling stride is set to $(1, 1, 2)$, which could reduce the number of bands. Then, feature maps in the shape of $(9 \times 9 \times 97, \ 24)$ are captured. After that, the dense spectral block combined by 3D-CNN with BN is attached. Each 3D-CNN of the dense spectral block has 12 channels with a $1 \times 1 \times 7$ kernel size. After attaching the dense spectral block, the channels of feature maps increase to 60 calculated by Equation (5). Therefore, we obtain feature maps with size of $(9 \times 9 \times 97, \ 60)$. Next, after the last 3D-CNN with kernel size of $1 \times 1 \times 97$, a $(9 \times 9 \times 1, \ 60)$ feature map is generated. However, the 60 channels make different contributions to the classification. To refine the spectral features, the channel attention block illustrated in Figure 4a and explained in Section 2.4.1 is adopted. The channel attention block reinforces the informative channels and whittles the information-lacking channels. After obtaining the weighted spectral feature maps by channel attention, a BN layer and a dropout layer are applied to enhance the numerical stability and vanquish the overfitting. Finally, via a global average pooling layer, the feature maps in the shape of $1 \times 60$ are obtained. The implementation of the spectral branch is available in Table 1.

**Table 1.** The implementation details of the spectral branch.

| Layer Name | Kernel Size | Output Size |
|:---:|:---:|:---:|
| Input | - | $(9 \times 9 \times 200)$ |
| Conv | $(1{\times}1{\times}7)$ | $(9 \times 9 \times 97, 24)$ |
| BN-Mish-Conv | $(1{\times}1{\times}7)$ | $(9 \times 9 \times 97, 12)$ |
| Concatenate | - | $(9 \times 9 \times 97, 36)$ |
| BN-Mish-Conv | $(1{\times}1{\times}7)$ | $(9 \times 9 \times 97, 12)$ |
| Concatenate | - | $(9 \times 9 \times 97, 48)$ |
| BN-Mish-Conv | $(1{\times}1{\times}7)$ | $(9 \times 9 \times 97, 12)$ |
| Concatenate | - | $(9 \times 9 \times 97, 60)$ |
| BN-Mish-Conv | $(1{\times}1{\times}97)$ | $(9 \times 9 \times 1, 60)$ |
| Channel Attention Block | - | $(9 \times 9 \times 1, 60)$ |
| BN-Dropout-GlobalAveragePooling | - | $(1 \times 60)$ |

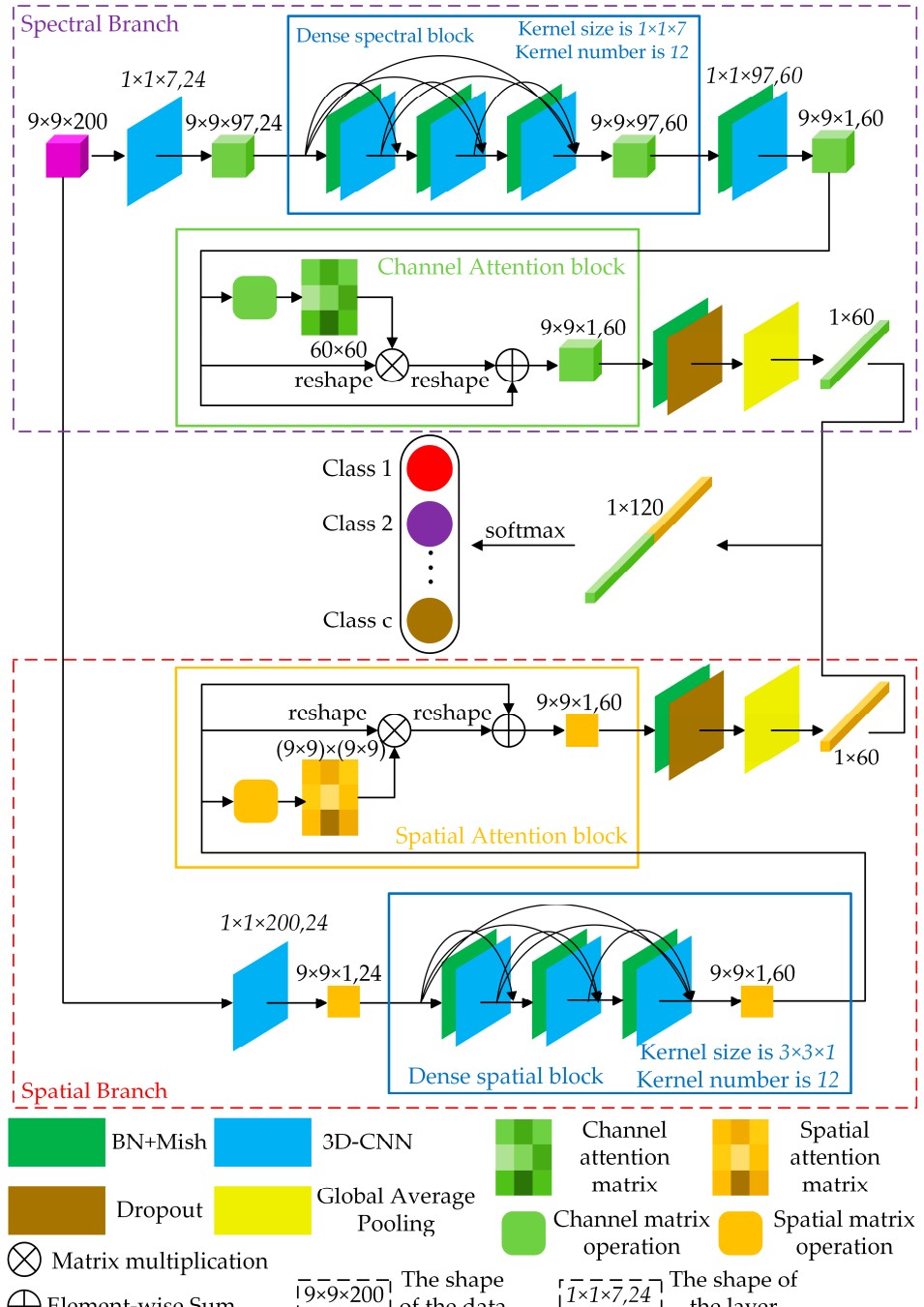

**Figure 6.** The structure of the DBDA network. The upper spectral branch composed of the dense spectral block and channel attention block is designed to capture spectral features. The lower spatial branch constituted by dense spatial block, and spatial attention block is designed to exploit spatial features.

### 3.1.2. Spatial Branch with the Spatial Attention Block

Meanwhile, the input data in the shape of $9 \times 9 \times 200$ are delivered to the spatial branch, and the initial 3D-CNN layer's size is set to $1 \times 1 \times 200$, which can compress spectral bands into one dimension. After that, feature maps in the shape of $(9 \times 9 \times 1, 24)$ are obtained. Then, the dense spatial block combined by 3D-CNN with BN is attached. Each 3D-CNN in the dense spectral block has 12 channels with a $3 \times 3 \times 1$ kernel size. Next, the extracted feature maps in the shape of $(9 \times 9 \times 1, 60)$ are fed into the spatial attention block, as illustrated in Figure 4b and expounded in Section 2.4.2. With the attention block, the coefficient of each pixel is weighted to get a more discriminative spatial feature.

After capturing the weighted spatial feature maps, a BN layer with a dropout layer is applied. Finally, the spatial feature maps in the shape of $1 \times 60$ are obtained via a global average pooling layer. The implementation of the spatial branch is given in Table 2.

**Table 2.** The implementation details of the spatial branch.

| Layer Name | Kernel Size | Output Size |
|---|---|---|
| Input | - | $(9 \times 9 \times 200)$ |
| Conv | $(1 \times 1 \times 200)$ | $(9 \times 9 \times 1, 24)$ |
| BN-Mish-Conv | $(3 \times 3 \times 1)$ | $(9 \times 9 \times 1, 12)$ |
| Concatenate | - | $(9 \times 9 \times 1, 36)$ |
| BN-Mish-Conv | $(3 \times 3 \times 1)$ | $(9 \times 9 \times 1, 12)$ |
| Concatenate | - | $(9 \times 9 \times 1, 48)$ |
| BN-Mish-Conv | $(3 \times 3 \times 1)$ | $(9 \times 9 \times 1, 12)$ |
| Concatenate | - | $(9 \times 9 \times 1, 60)$ |
| Channel Attention Block | - | $(9 \times 9 \times 1, 60)$ |
| BN-Dropout-GlobalAveragePooling | - | $(1 \times 60)$ |

### 3.1.3. Spectral and Spatial Fusion for HSI Classification

With the spectral branch and spatial branch, several spectral feature maps and spatial feature maps are obtained. Then, we perform a concatenation between two features for classification. Moreover, the reason why the concatenation operation is applied instead of add operation is that the spectral and spatial features are in the irrelevant domains, and the concatenate operation could keep them independent while the add operation would mix them together. In the end, the classification result is obtained via the fully connected layer and the softmax activation function.

For other datasets, network implementations are the same, and the only difference is the number of spectral bands. The whole methodology flowchart of DBDA is shown in Figure 7.

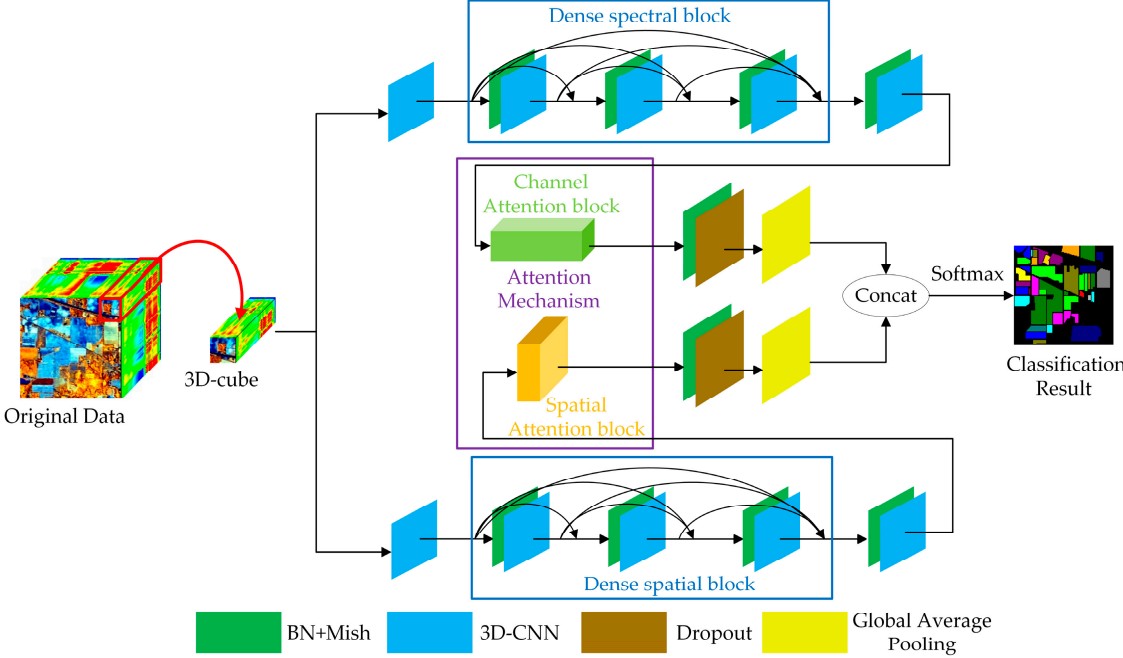

**Figure 7.** The flowchart for the DBDA methodology. The 3D-cube is fed into the spectral branch (top) and spatial branch (bottom) respectively. The obtained features are concatenated to classify the target pixel.

*3.2. Measures Taken to Prevent Overfitting*

Numerous training parameters and limited training samples cause the network to be prone overfitting. Thus, we take some measures to prevent overfitting.

3.2.1. A Strong and Appropriate Activation Function

The activation function brings the concept of nonlinearity to a neural network. An appropriate activation function can accelerate the speed of the counter-propagation and convergence of the network. The activation function we adopted is Mish [50], a self-regularized non-monotone activation function, instead of the conventional $ReLU(x) = max(0, x)$ [51]. The formula for the Mish is:

$$mish(x) = x \times tanh(softplus(x)) = x_i \times tanh(ln(1 + e^x)) \tag{11}$$

where $x$ represents the input of the activation. The comparison of Mish and ReLU can be seen in Figure 8. Mish is upper unbounded, and lower bounded with a scope of $[\approx -0.31, \infty)$. The differential coefficient definition of Mish is:

$$f'(x) = \frac{e^x \omega}{\delta^2} \tag{12}$$

where $\omega = 4(x + 1) + 4e^x + e^{3x} + e^x(4x + 6)$ and $\delta = 2e^x + e^{2x} + 2$.

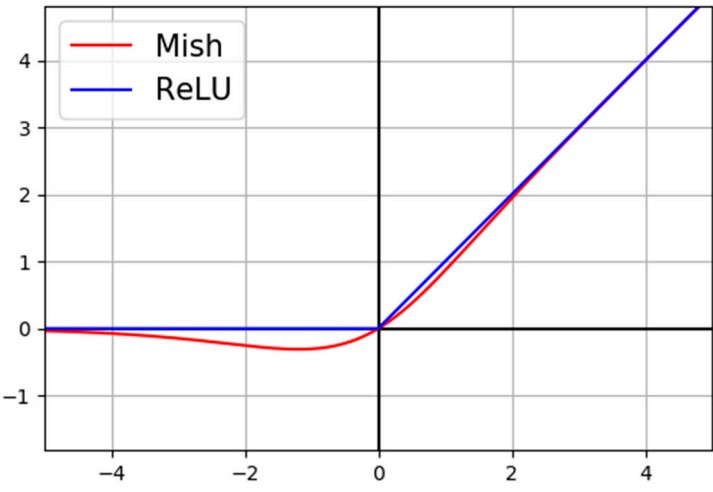

**Figure 8.** The graph of the activation functions (Mish and ReLU).

ReLU is a piecewise linear function that prunes all the negative inputs. Thus, if the input is nonpositive, then the neuron is going to "die" and cannot be activated anymore, even though negative inputs might contain useful information. On the contrary, negative inputs are preserved as negative outputs by Mish, which trades the input information and the network sparsity better.

3.2.2. Dropout Layer, Early Stopping Strategy and Dynamic Learning Rate Adjustment

A dropout layer [52] is adopted between the last BN layer and the global average pooling layer in the spatial branch and spectral branch separately. Dropout is a simple but effective method to prevent overfitting by dropping out units (hidden or visible) on a given percentage $p$ at the training phase. Moreover, the $p$ is selected as 0.5 in our framework. The existence of dropout makes the presence of other units unreliable, which prevents co-adaptation between units.

In addition, two training skills, the early stopping strategy, and the dynamic learning rate adjustment method are also introduced to our model. Early stopping signifies if the loss function is no longer decreasing for a certain number of epochs (the number is 20 in our model), then we would stop the training process early to prevent overfitting and reduce the training time.

The learning rate is a crucial hyper parameter to train a network, and dynamic learning rate can help a network avoid some local minima. The cosine annealing [53] method is adopted to adjust the learning rate dynamically as the following equation:

$$\eta_t = \eta_{min}^i + \frac{1}{2}\left(\eta_{max}^i - \eta_{min}^i\right)\left(1 + \cos\left(\frac{T_{cur}}{T_i}\pi\right)\right) \tag{13}$$

where $\eta_t$ is the learning rate within the $i$th run and $\left[\eta_{min}^i, \eta_{max}^i\right]$ is the range of the learning rate. $T_{cur}$ accounts for the count of epochs that have been executed, and $T_i$ controls the count of epochs that will be executed in a cycle of adjustment.

## 4. Experimental Results

To verify the accuracy and efficiency of the proposed model, experiments on four datasets are designed to compare and validate the accuracy and efficiency between the proposed network and other methods. The three quantitative metrics of overall accuracy (OA), average accuracy (AA), and Kappa coefficient (K) are used to measure the accuracy of each method. Concretely, OA represents the ratio of the true classifications of the entire pixels. AA means the average accuracy of all categories. The Kappa coefficient reflects the consistency between the ground truth and classification result. The higher the three metric values are, the better the classification result is. Meanwhile, we investigate the running time for each framework to evaluate its efficiency.

For each dataset, a certain number of training samples and validation samples are randomly selected from the labelled data on a certain percentage, and the rest of the samples are used to test the performance of the model. Since the proposed DBDA can maintain excellent performance when training samples are severely lacking, the amount of training samples and validation samples are set at a minimal level.

### 4.1. The Introduction about Datasets

In this paper, four widely used HSI datasets, the Indian Pines (IP) dataset, the Pavia University (UP) dataset, the Salinas Valley (SV) dataset, and the Botswana dataset (BS), are employed in the experiments.

**Indian Pines (IP):** Obtained through airborne visible infrared imaging spectrometer (AVIRIS) sensor in north-western Indiana, the Indian Pines dataset is composed of 200 spectral bands with a wavelength scope of 0.4 um to 2.5 um and 16 land cover classes. IP encompasses $145 \times 145$ pixels and owns the resolution of 20 m/pixel.

**Pavia University (UP):** Gathered by the reflective optics imaging spectrometer (ROSIS-3) sensor at the University of Pavia, northern Italy, the Pavia University dataset is composed of 103 spectral bands with a wavelength scope of 0.43 um to 0.86 um and 9 land cover classes. UP encompasses $610 \times 340$ pixels and owns the resolution of 1.3 m/pixel.

**Salinas Valley (SV):** Collected by the AVIRIS sensor from Salinas Valley, CA, USA, the Salinas Valley dataset is composed of 204 spectral bands with a wavelength scope of 0.4 um to 2.5 um and 16 land cover classes. SV encompasses $512 \times 217$ pixels and owns the resolution of 3.7 m/pixel.

**Botswana (BS):** Captured by the NASA EO-1 satellite over the Okavango Delta, Botswana, the Botswana dataset is composed of 145 spectral bands with a wavelength scope of 0.4 um to 2.5 um and 14 land cover classes. BS encompasses $1476 \times 256$ pixels and owns the resolution of 30 m/pixel.

Deep learning algorithms are data-driven, which rely on plenty of labelled training samples. The more labelled data are fed into training, the better accuracy is yielded. However, more data mean more time consumption and higher computation complexity. It is worth noting that the proposed DBDA can maintain excellent performance even though the training samples are very lacking. Therefore, the size of training samples and validation samples are set at a minimal level in the experiments. For IP, we select 3% samples for training, and 3% samples for validation. As the samples are enough for each class of UP and SV, we only select 0.5% samples for training, and 0.5% samples for validation. For

BS, the proportion of samples for training and validation is set to 1.2%. The reason why a decimal appears is that the number of samples in BS is small, so we set the ratio as 1% with a ceiling operation. Tables 3–6 list the samples of training, validation and testing for the four datasets.

**Table 3.** The samples for each category of training, validation and testing for the Indian Pines (IP) dataset.

| Order | Class | Total Number | Train | Val | Test |
|-------|-------|--------------|-------|-----|------|
| 1 | Alfalfa | 46 | 3 | 3 | 40 |
| 2 | Corn-notill | 1428 | 42 | 42 | 1344 |
| 3 | Corn-mintill | 830 | 24 | 24 | 782 |
| 4 | Corn | 237 | 7 | 7 | 223 |
| 5 | Grass-pasture | 483 | 14 | 14 | 455 |
| 6 | Grass-trees | 730 | 21 | 21 | 688 |
| 7 | Grass-pasture-mowed | 28 | 3 | 3 | 22 |
| 8 | Hay-windrowed | 478 | 14 | 14 | 450 |
| 9 | Oats | 20 | 3 | 3 | 14 |
| 10 | Soybean-notill | 972 | 29 | 29 | 914 |
| 11 | Soybean-mintill | 2455 | 73 | 73 | 2309 |
| 12 | Soybean-clean | 593 | 17 | 17 | 559 |
| 13 | Wheat | 205 | 6 | 6 | 193 |
| 14 | Woods | 1265 | 37 | 37 | 1191 |
| 15 | Buildings-Grass-Trees-Drives | 386 | 11 | 11 | 364 |
| 16 | Stone-Steel-Towers | 93 | 3 | 3 | 87 |
| | Total | 10,249 | 307 | 307 | 9635 |

**Table 4.** The samples for each category of training, validation and testing for the Pavia University (UP) dataset.

| Order | Class | Total Number | Train | Val | Test |
|-------|-------|--------------|-------|-----|------|
| 1 | Asphalt | 6631 | 33 | 33 | 6565 |
| 2 | Meadows | 18,649 | 93 | 93 | 18,463 |
| 3 | Gravel | 2099 | 10 | 10 | 2079 |
| 4 | Trees | 3064 | 15 | 15 | 3034 |
| 5 | Painted metal sheets | 1345 | 6 | 6 | 1333 |
| 6 | Bare Soil | 5029 | 25 | 25 | 4979 |
| 7 | Bitumen | 1330 | 6 | 6 | 1318 |
| 8 | Self-Blocking Bricks | 3682 | 18 | 18 | 3646 |
| 9 | Shadows | 947 | 4 | 4 | 939 |
| | Total | 42,776 | 210 | 210 | 42,356 |

*4.2. Experimental Setting*

To evaluate the effectiveness of DBDA, the deep-learning-based classifiers CDCNN [27], SSRN [31], FDSSC [32], and the state-of-the-art double-branch multi-attention mechanism network (DBMA) [34] are compared with our proposed framework. Furthermore, the SVM with RBF kernel [9] is also taken into account. The patch size of each classifier is set according to its original paper. To compare the training and testing consumptions of time, all experiments were executed on the same platform configured with 32 GB of memory and an NVIDIA GeForce RTX 2080Ti GPU. All deep-learning-based classifiers were implemented with PyTorch, and SVM was implemented with sklearn. Then, a brief introduction to the above methods will be given separately.

**Table 5.** The samples for each category of training, validation and testing for the Salinas Valley (SV) dataset.

| Order | Class | Total Number | Train | Val | Test |
|---|---|---|---|---|---|
| 1 | Brocoli-green-weeds-1 | 2009 | 10 | 10 | 1989 |
| 2 | Brocoli-green-weeds-2 | 3726 | 18 | 18 | 3690 |
| 3 | Fallow | 1976 | 9 | 9 | 1958 |
| 4 | Fallow-rough-plow | 1394 | 6 | 6 | 1382 |
| 5 | Fallow-smooth | 2678 | 13 | 13 | 2652 |
| 6 | Stubble | 3959 | 19 | 19 | 3921 |
| 7 | Celery | 3579 | 17 | 17 | 3545 |
| 8 | Grapes-untrained | 11,271 | 56 | 56 | 11,159 |
| 9 | Soil-vinyard-develop | 6203 | 31 | 31 | 6141 |
| 10 | Corn-senesced-green-weeds | 3278 | 16 | 16 | 3246 |
| 11 | Lettuce-romaine-4wk | 1068 | 5 | 5 | 1058 |
| 12 | Lettuce-romaine-5wk | 1927 | 9 | 94 | 1824 |
| 13 | Lettuce-romaine-6wk | 916 | 4 | 4 | 908 |
| 14 | Lettuce-romaine-7wk | 1070 | 5 | 5 | 1060 |
| 15 | Vinyard-untrained | 7268 | 36 | 36 | 7196 |
| 16 | Vinyard-vertical-trellis | 1807 | 9 | 9 | 1789 |
| | Total | 54,129 | 263 | 348 | 53,603 |

**Table 6.** The samples for each category of training, validation and testing for the Botswana dataset (BS) dataset.

| Order | Class | Total Number | Train | Val | Test |
|---|---|---|---|---|---|
| 1 | Water | 270 | 3 | 3 | 264 |
| 2 | Hippo grass | 101 | 2 | 2 | 97 |
| 3 | Floodplain grasses1 | 251 | 3 | 3 | 245 |
| 4 | Floodplain grasses2 | 215 | 3 | 3 | 209 |
| 5 | Reeds1 | 269 | 3 | 3 | 263 |
| 6 | Riparian | 269 | 3 | 3 | 263 |
| 7 | Fierscar2 | 259 | 3 | 3 | 253 |
| 8 | Island interior | 203 | 3 | 3 | 197 |
| 9 | Acacia woodlands | 314 | 4 | 4 | 306 |
| 10 | Acacia shrublands | 248 | 3 | 3 | 242 |
| 11 | Acacia grasslands | 305 | 4 | 4 | 297 |
| 12 | Short mopane | 181 | 2 | 2 | 177 |
| 13 | Mixed mopane | 268 | 3 | 3 | 262 |
| 14 | Exposed soils | 95 | 1 | 1 | 93 |
| | Total | 3248 | 40 | 40 | 3168 |

**SVM:** For SVM with a radial basis function (RBF) kernel, all individual pixels with their spectral bands are fed in directly.

**CDCNN:** The architecture of the CDCNN is shown in [27], which is based on 2D-CNN and ResNet. The size of input is $5 \times 5 \times b$, where $b$ denotes the number of spectral bands.

**SSRN:** The architecture of the SSRN is proposed in [31], which is based on 3D-CNN and ResNet. The size of the input is $7 \times 7 \times b$.

**FDSSC:** The architecture of the FDSSC can be seen in [32], which is based on 3D-CNN and DenseNet. The size of the input is $9 \times 9 \times b$.

**DBMA:** The architecture of the DBMA is presented in [34], which is based on 3D-CNN, DenseNet, and an attention mechanism. $7 \times 7 \times b$ is the input patch size.

For CDCNN, SSRN, FDSSC, DBMA, and the proposed method, the batch size is set as 16, and the optimizer is set to Adam with the 0.0005 learning rate. The upper limit of the early stopping strategy

is set to 200 epochs. If the loss in the validation set no longer declines for 20 epochs, then we would terminate the training phase.

### 4.3. Classification Maps and Categorized Results

#### 4.3.1. Classification Maps and Categorized Results for the IP Dataset

The categorized results using different methods for the IP dataset are demonstrated in Table 7 where the best class-specific accuracy is in bold, and classification maps of the different methods and ground truth are shown in Figure 9.

**Table 7.** The categorized results for the IP dataset with 3% training samples.

| Class | Color | SVM | CDCNN | SSRN | FDSSC | DBMA | Proposed |
|-------|-------|------|-------|-------|-------|-------|----------|
| 1 | | 24.24 | 0.00 | **100.0** | 85.42 | 93.48 | **100.0** |
| 2 | | 58.10 | 62.36 | 89.14 | 97.20 | 91.15 | 88.49 |
| 3 | | 64.37 | 57.00 | 77.49 | 94.45 | 99.58 | 97.12 |
| 4 | | 37.07 | 37.50 | 88.95 | **100.0** | 98.57 | **100.0** |
| 5 | | 87.67 | 88.16 | 96.48 | **100.0** | 97.45 | **100.0** |
| 6 | | 84.02 | 79.63 | 98.15 | **100.0** | 95.66 | 97.18 |
| 7 | | 56.10 | 0.00 | 0.00 | 73.53 | 40.00 | 92.59 |
| 8 | | 89.62 | 84.02 | 84.54 | 99.78 | **100.0** | 99.78 |
| 9 | | 21.21 | 0.00 | 0.00 | **100.0** | 38.10 | **100.0** |
| 10 | | 65.89 | 37.50 | 92.07 | 89.25 | 85.98 | 89.87 |
| 11 | | 62.32 | 53.25 | 90.89 | 93.97 | 94.39 | 99.33 |
| 12 | | 52.40 | 42.96 | 84.19 | 95.41 | 89.92 | 98.50 |
| 13 | | 94.30 | 49.47 | 98.47 | **100.0** | 99.48 | 96.02 |
| 14 | | 90.15 | 76.71 | 94.56 | 93.14 | 92.81 | 93.22 |
| 15 | | 63.96 | 62.60 | 84.11 | 90.61 | 89.66 | 96.99 |
| 16 | | 98.46 | 83.70 | 91.40 | 96.55 | 96.55 | 94.38 |
| OA | | 69.41 | 62.32 | 89.81 | 94.87 | 93.15 | 95.38 |
| AA | | 65.62 | 50.93 | 79.40 | 94.33 | 87.67 | 96.47 |
| kappa | | 0.6472 | 0.5593 | 0.8839 | 0.9414 | 0.9219 | 0.9474 |

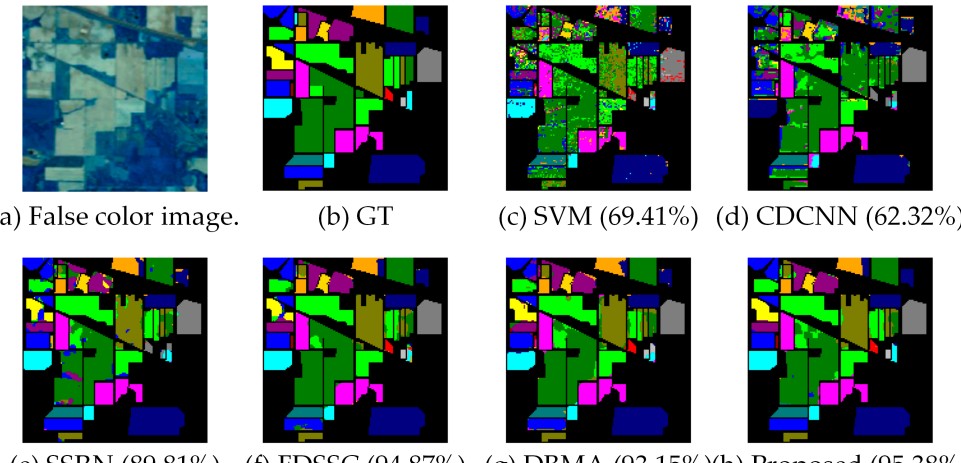

(a) False color image.    (b) GT    (c) SVM (69.41%)    (d) CDCNN (62.32%)

(e) SSRN (89.81%)    (f) FDSSC (94.87%)    (g) DBMA (93.15%)(h) Proposed (95.38%)

**Figure 9.** Classification maps for the IP dataset using 3% training samples. (**a**) False-color image. (**b**) Ground-truth (GT). (**c**–**h**) The classification maps with disparate algorithms.

Our proposed framework obtains the best results with 95.38% OA, 96.47% AA, and 0.9474 Kappa, which can be seen from Table 7.CDCNN based on 2D-CNN achieves the worst accuracy with 62.32%

OA, due to the limited training samples and weak network structure. Although SVM performs better than CDCNN with more than 7% in OA, the salt-and-pepper noise is severe, which can be seen in Figure 9c. Because SVM uses no spatial neighborhood information. The 3D-CNN based models far exceed SVM and CDCNN, owing to its incorporation of both spatial and spectral information in the classification. FDSSC uses dense connection instead of residual connection, which enhances the performance of the network and obtains more than 5% improvement in OA compared to SSRN. Based on FDSSC, DBMA extracts the spatial and spectral features in two independent branches and brings the attention mechanism in. However, when training samples are very lacking, DBMA might overfit the training data. With our proposed framework DBDA, it can accomplish stable and reliable performance with limited data duo to its flexible and adaptive attention mechanism, the appropriate activation function, and the other measures to prevent overfitting.

Taking class 7, which only has three training samples in the IP dataset, as an example, our method performs well and obtains an acceptable consequence of 92.59%, while the results of other methods (SVM: 56.10%, CDCNN: 0.00%, SSRN: 0.00%, FDSSC: 73.53%, and DBMA: 40.00%) are not very satisfactory.

Overall, the proposed model improves the OA by 2.23%, the AA by 8.80%, and the kappa by 0.0225 compared to DBMA.

### 4.3.2. Classification Maps and Categorized Result for the UP Dataset

The categorized results using different methods for the UP dataset are demonstrated in Table 8 where the best class-specific accuracy is in bold, and classification maps for the different methods and ground truth are shown in Figure 10.

We can see that our proposed method obtains the best results regarding the three indexes fromTable 8. Though our method cannot make every class precision best, the accuracy of each class using our method exceeds 89%, which means our method is able to capture the distinctive features between different classes.

Since the samples in the UP dataset are sufficient, there are enough samples for each class even if we just choose 0.5% training samples. Thus, DBMA overcomes overfitting and performs better than FDSSC because of its superior architecture. CDCNN with ample samples surpasses the performance of SVM.

**Table 8.** The categorized results for the UP dataset with 0.5% training samples.

| Class | Color | SVM | CDCNN | SSRN | FDSSC | DBMA | Proposed |
|---|---|---|---|---|---|---|---|
| 1 | | 82.87 | 85.74 | 99.15 | 96.88 | 94.86 | 89.03 |
| 2 | | 88.07 | 94.45 | 98.06 | 97.57 | 96.57 | 98.32 |
| 3 | | 70.84 | 32.59 | 96.64 | 89.97 | **100.0** | 98.70 |
| 4 | | 95.61 | 97.46 | 99.86 | 99.21 | 97.44 | 98.42 |
| 5 | | 92.24 | 99.10 | 99.85 | 99.55 | 95.69 | 99.78 |
| 6 | | 76.98 | 80.88 | 96.88 | 97.97 | 96.78 | 98.57 |
| 7 | | 68.98 | 88.83 | 73.24 | **100.0** | 95.69 | 95.84 |
| 8 | | 71.14 | 66.19 | 82.36 | 70.97 | 78.93 | 89.47 |
| 9 | | 99.89 | 96.01 | **100.0** | **100.0** | 99.55 | 99.89 |
| OA | | 84.29 | 87.70 | 95.59 | 94.43 | 94.72 | 96.00 |
| AA | | 82.96 | 82.36 | 94.01 | 94.68 | 95.49 | 96.45 |
| kappa | | 0.7883 | 0.8359 | 0.9415 | 0.9257 | 0.9295 | 0.9467 |

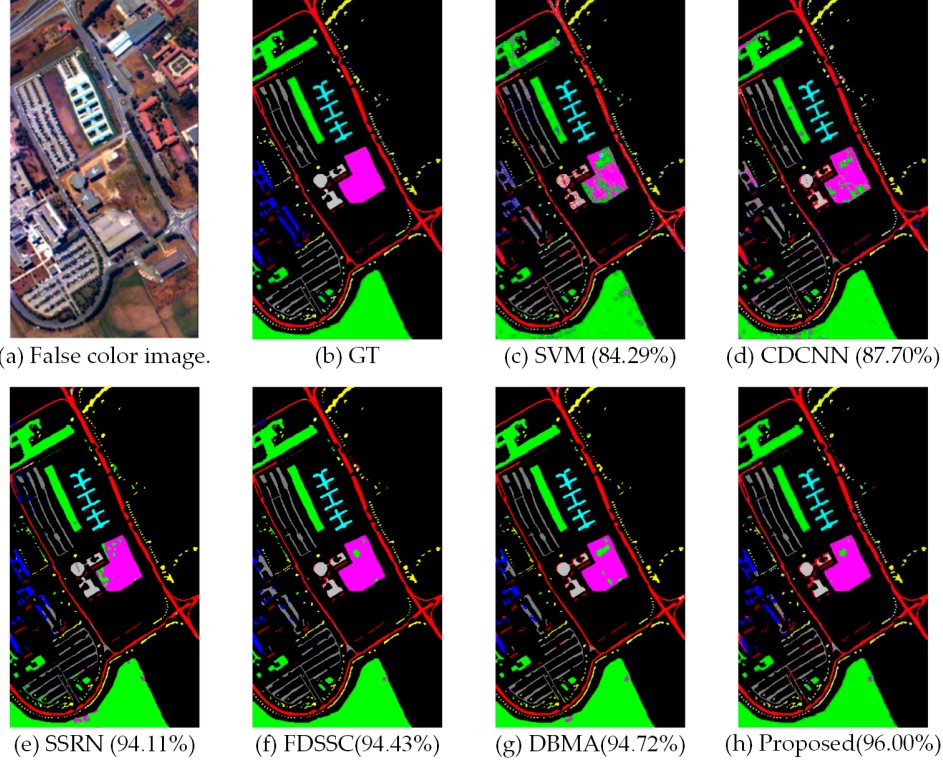

(a) False color image.     (b) GT     (c) SVM (84.29%)     (d) CDCNN (87.70%)

(e) SSRN (94.11%)     (f) FDSSC(94.43%)     (g) DBMA(94.72%)     (h) Proposed(96.00%)

**Figure 10.** Classification maps for the UP dataset using 0.5% training samples. (**a**) False-color image. (**b**) Ground-truth (GT). (**c–h**) The classification maps with disparate algorithms.

### 4.3.3. Classification Maps and Categorized Results for the SV Dataset

The categorized results using the different methods for the SV dataset are demonstrated in Table 9 where the best class-specific accuracy is in bold, and classification maps of the different methods and ground truth are shown in Figure 11.

**Table 9.** The categorized results for the SV dataset with 0.5% training samples.

| Class | Color | SVM | CDCNN | SSRN | FDSSC | DBMA | Proposed |
|-------|-------|------|--------|--------|--------|--------|----------|
| 1 |  | 99.85 | 0.00 | **100.0** | **100.0** | **100.0** | **100.0** |
| 2 |  | 98.95 | 64.82 | **100.0** | **100.0** | 99.51 | 99.17 |
| 3 |  | 89.88 | 94.69 | 89.72 | 99.44 | 98.92 | 97.74 |
| 4 |  | 97.30 | 82.99 | 94.85 | 98.57 | 96.39 | 95.95 |
| 5 |  | 93.56 | 98.24 | 99.39 | 99.87 | 96.39 | 99.29 |
| 6 |  | 99.89 | 96.51 | 99.95 | 99.97 | 99.17 | 99.92 |
| 7 |  | 91.33 | 95.98 | 99.75 | 99.75 | 96.80 | 99.83 |
| 8 |  | 74.73 | 88.23 | 88.60 | 99.60 | 95.60 | 95.97 |
| 9 |  | 97.69 | 99.26 | 98.48 | 99.69 | 99.22 | 99.37 |
| 10 |  | 90.01 | 67.39 | 98.81 | 99.02 | 96.20 | 96.72 |
| 11 |  | 75.92 | 72.03 | 93.30 | 92.77 | 82.29 | 93.72 |
| 12 |  | 95.19 | 75.49 | 99.95 | 99.64 | 99.17 | **100.0** |
| 13 |  | 94.87 | 95.71 | **100.0** | **100.0** | 98.91 | **100.0** |
| 14 |  | 89.26 | 94.92 | 97.86 | 98.05 | 98.22 | 96.89 |
| 15 |  | 75.86 | 51.88 | 89.96 | 74.58 | 84.71 | 93.42 |
| 16 |  | 99.03 | 99.62 | **100.0** | **100.0** | **100.0** | **100.0** |
| OA |  | 88.09 | 77.79 | 94.72 | 94.99 | 95.44 | 97.51 |
| AA |  | 91.45 | 79.86 | 96.66 | 97.56 | 96.34 | 98.00 |
| kappa |  | 0.8671 | 0.7547 | 0.9412 | 0.9444 | 0.9493 | 0.9723 |

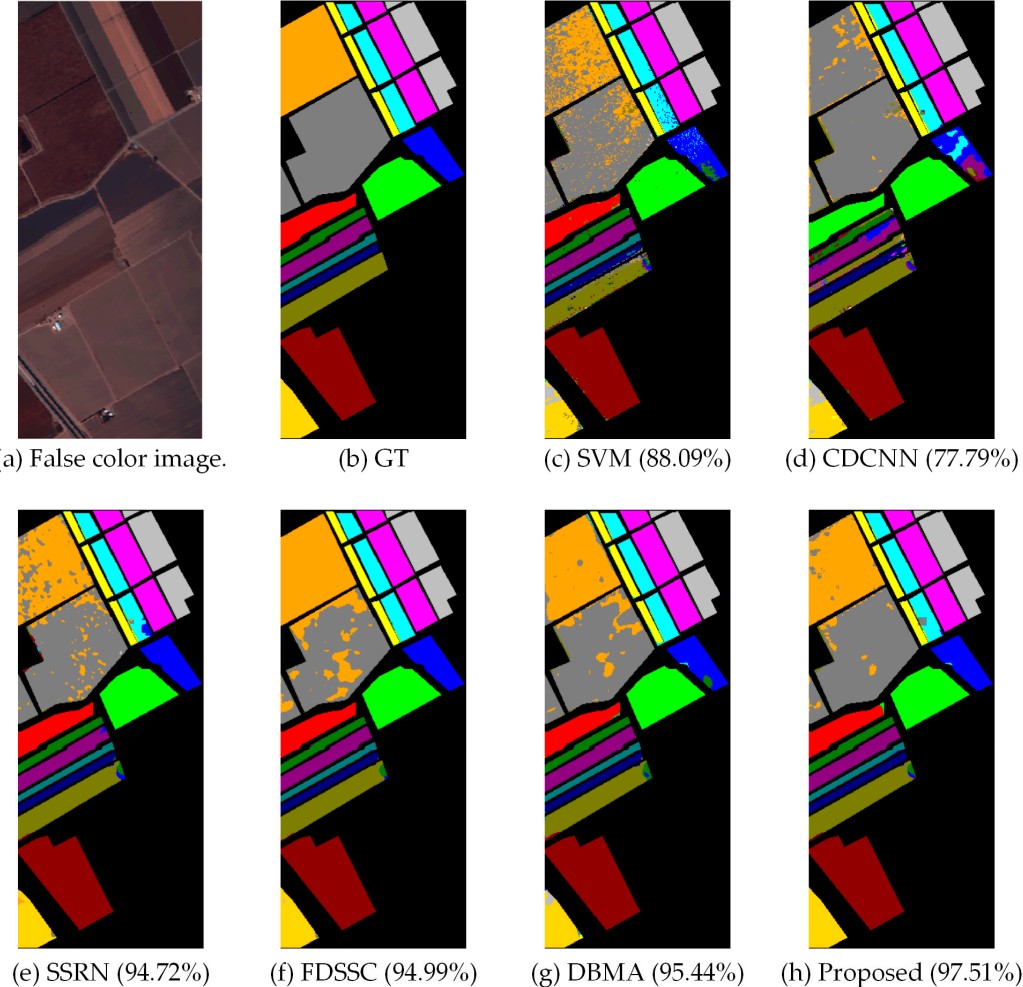

(a) False color image.      (b) GT      (c) SVM (88.09%)      (d) CDCNN (77.79%)

(e) SSRN (94.72%)      (f) FDSSC (94.99%)      (g) DBMA (95.44%)      (h) Proposed (97.51%)

**Figure 11.** Classification maps for the UP dataset using 0.5% training samples. (**a**) False-color image. (**b**) Ground-truth (GT). (**c–h**) The classification maps with disparate algorithms.

We can see that our proposed method obtains the best results regarding the three indexes from Table 9, and the accuracy of each category classified by our method exceeds 93%.

Similarly, because of the sufficient samples in the SV dataset, 0.5% training samples are enough. Thus, DBMA once again performs better than FDSSC. However, the SV dataset owns 16 classes while the UP dataset only has 9 classes, so CDCNN obtains a weaker performance than SVM.

### 4.3.4. Classification Maps and Categorized Result for the BS Dataset

The categorized results using different methods for the BS dataset are demonstrated in Table 10 where the best class-specific accuracy is in bold, and classification maps of the different methods and ground truth are shown in Figure 12.

Since the BS dataset is small and only with 3, 248 labelled samples, just 40 samples are selected as the training set and 40 samples are chosen as the validation set. Nonetheless, our method achieves 96.24% OA performance, 2.81% higher than DBMA. One reason is that our method can capture spatial and spectral features more effectively.

**Table 10.** The categorized results for the BS dataset with 1.2% training samples.

| Class | Color | SVM | CDCNN | SSRN | FDSSC | DBMA | Proposed |
|-------|-------|-----|-------|------|-------|------|----------|
| 1 | | **100.0** | 94.60 | 94.95 | 97.41 | 97.77 | 95.64 |
| 2 | | 97.56 | 68.64 | **100.0** | 98.95 | 88.89 | 98.99 |
| 3 | | 86.35 | 81.11 | 91.42 | **100.0** | **100.0** | **100.0** |
| 4 | | 63.51 | 65.45 | 97.34 | 93.03 | 92.51 | 91.30 |
| 5 | | 84.33 | 89.10 | 92.42 | 80.74 | 93.51 | 95.58 |
| 6 | | 61.27 | 69.28 | 66.39 | 84.93 | 68.94 | 82.23 |
| 7 | | 82.09 | 80.07 | **100.0** | 84.62 | **100.0** | **100.0** |
| 8 | | 63.46 | 89.36 | **100.0** | 93.36 | 96.10 | 95.63 |
| 9 | | 63.53 | 55.53 | 90.75 | 88.44 | 85.15 | 96.50 |
| 10 | | 65.74 | 81.69 | 86.83 | 99.59 | 97.60 | 98.79 |
| 11 | | 93.91 | 92.48 | **100.0** | 99.67 | 99.66 | 99.67 |
| 12 | | 90.70 | 90.91 | **100.0** | **100.0** | 97.79 | **100.0** |
| 13 | | 73.62 | 88.59 | 94.83 | 81.59 | **100.0** | **100.0** |
| 14 | | 92.98 | **100.0** | **100.0** | **100.0** | **100.0** | **100.0** |
| OA | | 77.21 | 80.90 | 91.89 | 91.57 | 93.43 | 96.24 |
| AA | | 79.93 | 81.92 | 93.92 | 93.02 | 94.14 | 96.74 |
| kappa | | 0.7532 | 0.7930 | 0.9121 | 0.9086 | 0.9.89 | 0.9593 |

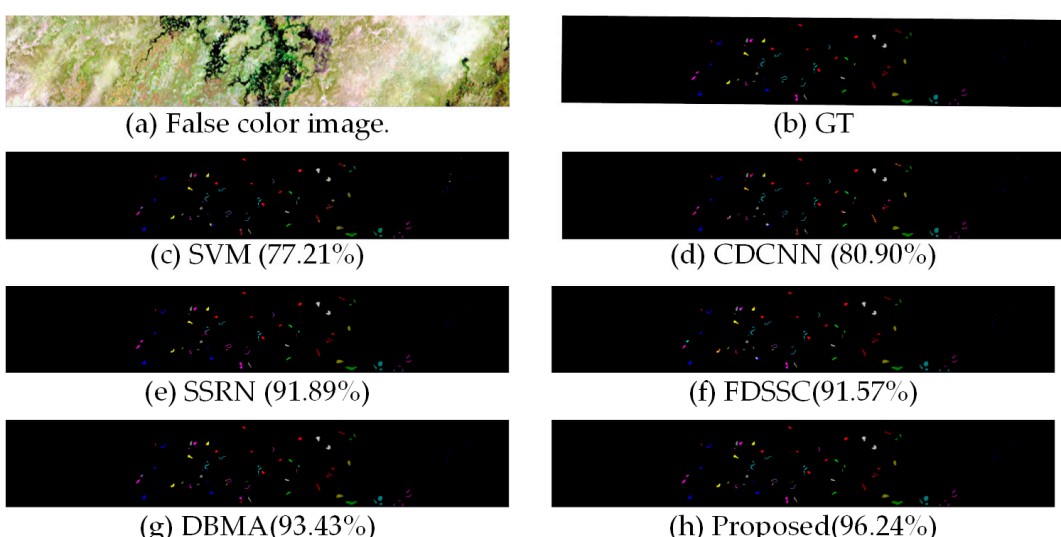

(a) False color image.

(b) GT

(c) SVM (77.21%)

(d) CDCNN (80.90%)

(e) SSRN (91.89%)

(f) FDSSC(91.57%)

(g) DBMA(93.43%)

(h) Proposed(96.24%)

**Figure 12.** Classification maps for the BS dataset using 1.2% training samples. (**a**) False-color image. (**b**) Ground-truth (GT). (**c–h**) The classification maps with disparate algorithms.

### 4.4. Investigation of Running Time

The above experiments prove that our proposed method can achieve a higher degree of accuracy with less data. However, a good method should balance the accuracy and efficiency properly. This part is executed to measure the efficiency of each method. Tables 11–14 list the consumptions of time for the six algorithms on the IP, UP, SV, and BS datasets.

Since we use SVM as a pixel-based model, it spends less time than 3D-cube-based models in most cases. On account of 2D-CNN containing less parameters to be trained, CDCNN takes less time than 3D-CNN-based models.

**Table 11.** Training and testing consumption of support vector machines (SVM), contextual deep convolutional neural networks (CDCNN), spectral-spatial residual network (SSRN), fast dense spectral-spatial convolution (FDSSC), double-branch multi-attention (DBMA), and our method on the IP dataset using 307 training samples (3%) in 16 classes.

| Dataset | Algorithm | Training Times (s) | Testing Times (s) |
|---------|-----------|--------------------|-------------------|
| | SVM | 20.10 | 0.66 |
| | CDCNN | 11.13 | 1.54 |
| **Indian Pines** | SSRN | 46.03 | 2.71 |
| | FDSSC | 105.05 | 4.86 |
| | DBMA | 94.69 | 6.35 |
| | Proposed | 69.83 | 5.60 |

**Table 12.** Training and testing consumption of SVM, CDCNN, SSRN, FDSSC, DBMA, and our method on the UP dataset using 210 training samples (0.5%) in nine classes.

| Dataset | Algorithms | Training Times (s) | Testing Times (s) |
|---------|-----------|--------------------|-------------------|
| | SVM | 3.38 | 2.29 |
| | CDCNN | 10.26 | 4.92 |
| **Pavia University** | SSRN | 9.93 | 6.41 |
| | FDSSC | 26.01 | 11.56 |
| | DBMA | 21.02 | 11.17 |
| | Proposed | 18.46 | 13.32 |

**Table 13.** Training and testing consumption of SVM, CDCNN, SSRN, FDSSC, DBMA, and our method on the SV dataset using 263 training samples (0.5%) in 16 classes.

| Dataset | Algorithms | Training Times (s) | Testing Times (s) |
|---------|-----------|--------------------|-------------------|
| | SVM | 9.35 | 3.89 |
| | CDCNN | 9.82 | 6.14 |
| **Salinas** | SSRN | 73.75 | 13.99 |
| | FDSSC | 99.91 | 25.57 |
| | DBMA | 105.30 | 31.82 |
| | Proposed | 71.18 | 23.93 |

**Table 14.** Training and testing consumption of SVM, CDCNN, SSRN, FDSSC, DBMA, and our method on the BS dataset using 40 training samples (1.2%) in 14 classes.

| Dataset | Algorithms | Training Times (s) | Testing Times (s) |
|---------|-----------|--------------------|-------------------|
| | SVM | 0.93 | 0.15 |
| | CDCNN | 11.10 | 1.33 |
| **Botswana** | SSRN | 8.87 | 1.37 |
| | FDSSC | 17.84 | 1.45 |
| | DBMA | 13.67 | 2.04 |
| | Proposed | 17.19 | 1.90 |

For 3D-CNN-based models, the proposed method consumes less training time compared to FDSSC and DBMA while obtaining better performance because of its higher rate of convergence. Even though SSRN is quicker than our method, the accuracy of our method is superior. That is, our method can balance the accuracy and efficiency better.

## 5. Discussion

In this part, further assessments of DBDA are conducted. First, different proportions of training samples are fed into the network, and the results reflect that our method can maintain effectiveness especially when the training samples are severely limited. Second, the results of ablation experiments

confirm the necessity of the attention mechanism. Third, the results of the different activation functions show that Mish is a better choice than ReLU for DBDA.

### 5.1. Investigation of the Proportion of Training Samples

As we mentioned, deep learning is a data-driven algorithm that depends on large amounts of high-quality labelled dataset. In this part, we investigate the scenarios for different proportions of training samples.

Figure 13 demonstrates the experimental results. For the IP and BS datasets, we use 0.5%, 1%, 3%, 5%, and 10% samples as the training sets, respectively. For the UP and SV datasets, we use 0.1%, 0.5%, 1%, 5%, and 10% of samples as the training sets, respectively.

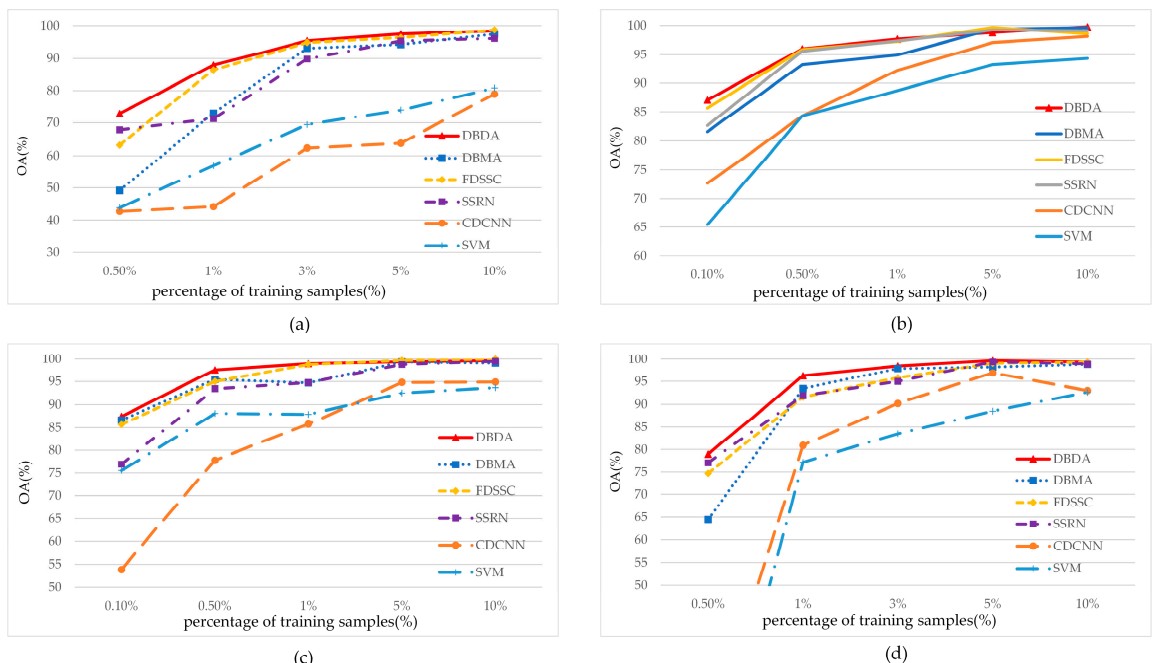

**Figure 13.** The OA results of SVM, CDCNN, CDCNN, SSRN, FDSSC, DBMA and our proposed method with varying proportions of training samples on the (**a**) IP, (**b**) UP, (**c**) SV and (**d**) BS.

As we expected, the accuracy improves with increase in the number of training samples. All 3D-based methods, including SSRN, FDSSC, DBMA, and the proposed framework can obtain near-perfect performances as long as enough samples (about 10% of the whole dataset) are provided. At the same time, the performance gaps between different models are narrowed according to the increases in training samples. Nevertheless, our method outpaces other methods, especially when samples are insufficient. Since it is costly to label the dataset, our proposed method can save labor and cost.

### 5.2. Effectiveness of the Attention Mechanism

To verify the effectiveness of the attention mechanism, we remove the spatial-attention module, spectral-attention module, and both attention modules of the DBDA respectively, and compare the performance between these three "incomplete DBDA" and the "complete DBDA".

From Figure 14, we can conclude that the existence of the spatial attention mechanism and the spectral attention mechanism does promote the accuracy on four datasets.

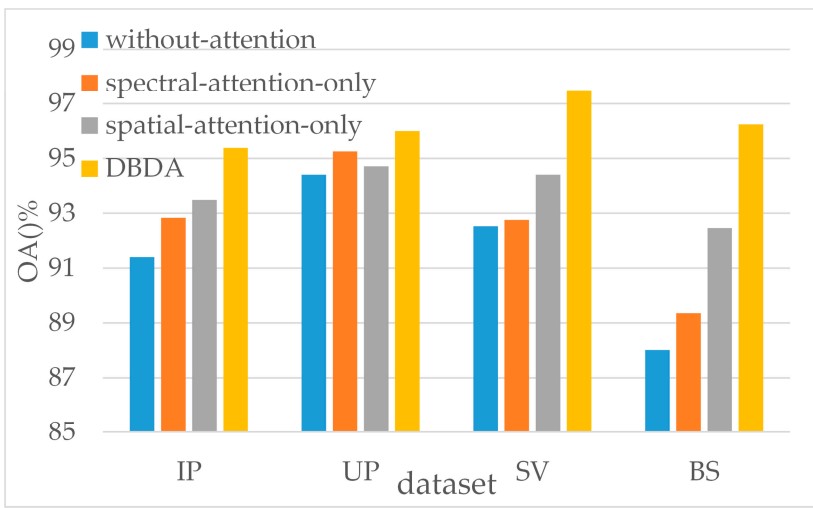

**Figure 14.** Effectiveness of the attention mechanism (results of different attention mechanisms).

Averagely, the attention mechanism improves 4.69% OA on four datasets. Furthermore, a single spatial attention mechanism (average 2.18% improvement) performs better than a single spectral attention mechanism (average 0.97% improvement) upon most occasions.

*5.3. Effectiveness of the Activation Function*

In Section 3.2.1, we illustrate why we adopted Mish as the activation function rather than the generally used ReLU. Here, we will compare the performance between DBDA based on Mish and DBDA based on ReLU. Figure 15 shows the classification OA of them.

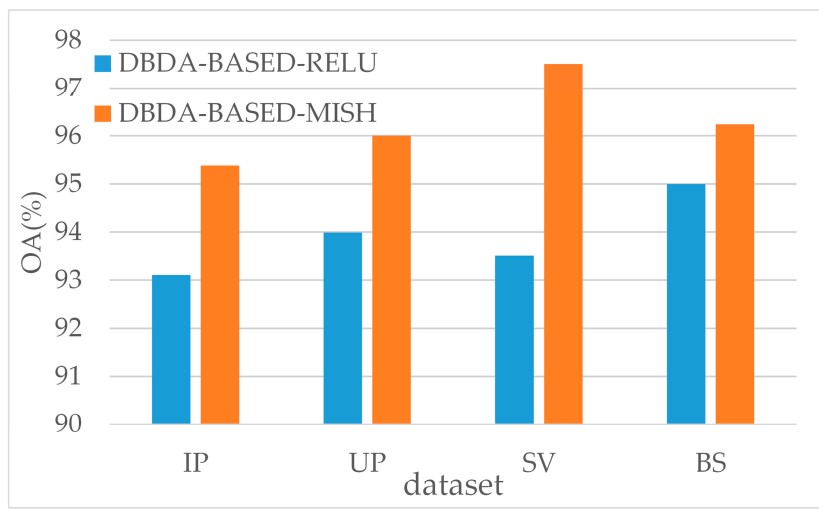

**Figure 15.** Effectiveness of the activation function (results on different activation functions).

As shown in Figure 15, DBDA based on Mish surpasses DBDA based on ReLU. Specifically, there are 2.27%, 2.01%, 4.00% and 1.24% OA improvements on the IP, UP, SV, and BS datasets, respectively. Since Mish can quicken counter-propagation, the difference in performance occurs.

## 6. Conclusions

In this paper, we proposed an end-to-end framework double-branch dual-attention mechanism network for HSI classification. The input of the DBDA framework is original 3D pixel data without any cumbersome pre-processing to reduce dimensionality. Based on densely connected 3D-CNN layers

with BN, we designed two branches that capture spectral and spatial features respectively. Meanwhile, a flexible and adaptive self-attention mechanism was applied to spectral branch and spatial branch, respectively. Mish was introduced as the activation function to accelerate the counter-propagation and convergence processes. Dynamic learning rates, early stopping, and dropout layers were also adopted to prevent overfitting.

Extensive experimental results demonstrate that our proposed framework surpasses the state-of-the-art algorithm, especially when training samples are finite and limited. Meanwhile, the consumption of time is also decreased in comparison to FDSSC and DBMA, as the attention blocks and the activation function Mish accelerate the convergent speed of the model. Accordingly, we draw a conclusion that the structure of our method is more preferable for HSI classification.

A future direction of our work is applying our proposed framework to other hyperspectral images, not just process the above-mentioned open-source datasets. Moreover, it is also an attractive challenge to reduce the training time.

**Author Contributions:** Conceptualization, R.L.; formal analysis, R.L.; funding acquisition, S.Z.; methodology, R.L.; validation, R.L.; writing—original draft, R.L. and C.D.; writing—review and editing, S.Z., C.D., Y.Y., and X.W. All authors have read and agreed to the published version of the manuscript.

**Funding:** This research was funded by the National Natural Science Foundations of China (No. 41671452).

**Acknowledgments:** I am indebted to my mentor, Shunyi Zheng, who supported my work strongly. I would also like to express my gratitude to Chenxi Duan, my morning sunlight who revised my manuscript earnestly.

**Conflicts of Interest:** The authors declare no conflicts of interest. The founding sponsors had no role in the design of the study; in the collection, analyses, or interpretation of data; in the writing of the manuscript; nor in the decision to publish the results.

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
