# Peer review of "Classification of Hyperspectral Image Based on Double-Branch Dual-Attention Mechanism Network"

_remotesensing, doi:10.3390/rs12030582_

Round 1

Reviewer 1 Report

Dear authors. In my opinion, the paper is not well written.  More importantly, to introduce a new methodology (Classification of Hyperspectral Image Based on Double-Branch Dual-Attention Mechanism Network ) a suitable experimental design should be used.  This would require to have some very clear objectives. In my  opinion, objectives are not clear, and the results and validation sections are not convincing ... and generally poorly illustrated. Therefore, for the  time being, I cannot recommend to publish your manuscript. It need a major revision.

 Re-write the abstract, introduction and method sections. 

The method section is weak. You need to clarify the four steps (add a methodology flowchart, etc.)

all Figures are too small and hard to see clearly! Make larger please. (Legend is not clear).

Figures and tables caption should be complete to let readers understand it well without referring to the text.

Overall, English grammar and formations are still poor, and need to be improved.

I personally think that your conclusion are weak.

Reviewer 2 Report

I have read the paper entitled "Classification of Hyperspectral Image Based on Double-Branch Dual-Attention Mechanism Network". In the article it is faced the problem of Hyperspectral Image (HSI) classification. To face the HSI problem, the authors design a deep learning model named "Double-Branch Dual-Attention mechanism network (DBDA)" which takes inspiration by DenseNet. The work is well presented and any point is analysed in the deep: 1) the introduction and related works sections are wide. 2) The framework section is presented in the details 3) the experimental section is convincing and there is a good comparison with the state of the arts method.

The article is of great quality and I recommend its publication without any doubt.

Reviewer 3 Report

The strategy proposed by the authors is interesting and original, and results are encouraging. However, there are several major issues that need to be addressed, especially concerning language and the description of the experimental setup, as detailed below.
- Language needs a lot of work. There are many sentences that are borderline nonsensical, and grammatical errors are aplenty. A full revision by someone more knowledgeable on the English language is highly recommended.
- Page 2: please provide some examples of studies applying SAE, DAE, RAE, DBN, CNN, RNN, and GAN to HSI problems.
- There are a number of references with the error “Error! Reference source not found.” Please check.
- Section 2.4.1, 1st paragraph: please provide a reference and a brief description of the so-called “Indian Pines” dataset. This is done later (Section 3), but since the dataset is first introduced at this point, it needs to be defined here. The authors should also explain what each dimension represents in the matrices mentioned in this section (e.g. 9×9×97,24). Without this information it is difficult to understand the strategy proposed by the authors.
- Section 2 as a whole is difficult to follow. This is partly due to the language problems mentioned above, but the organization of the section could also be improved.
- Section 3.1: the four datasets are described here, but the authors failed to mention how many samples are present in each set. Once again, the relevant information is presented later in the text, but the number of samples should be mentioned here; my suggestion would be referencing Tables 3 to 6 here.
- The “experimental setup” description is confusing and incomplete, leading to several unanswered questions. The spatial dimensions of each dataset are different, and each classifier also receives inputs of different sizes. How were the original images processed prior to being fed to the classifiers? Were they downsampled or subdivided into patches with the desired dimensions? If the later strategy was used, was there some selection of the regions of the interest in order to avoid feeding the classifiers with irrelevant data? It is also not clear why the authors chose to use so few samples for training and validation. Sure, training deep learning models is a computationally intensive task, but the number of images is not that high… unless they were subdivided in several small patches, but this is not clear. This section needs to be extensively revised for the sake of clarity.
- Ablation experiments need to be better explained.

Round 2

Reviewer 1 Report

Now it's okay. 

Reviewer 3 Report

My concerns and suggestions were properly addressed. My only suggestion for the authors at this point is to carefully revise the text to correct the few grammatical errors and typos that still remain.
